# Extended haplotype-phasing of long-read de novo genome assemblies using Hi-C

Zev N. Kronenberg 1,2✉, Arang Rhie3, Sergey Koren 3, Gregory T. Concepcion 2, Paul Peluso2, Katherine M. Munson 4, David Porubsky4, Kristen Kuhn5, Kathryn A. Mueller1, Wai Yee Low 6, Stefan Hiendleder 6, Olivier Fedrigo7, Ivan Liachko1, Richard J. Hall 2, Adam M. Phillippy 3, Evan E. Eichler4,8, John L. Williams 6,9, Timothy P. L. Smith 5, Erich D. Jarvis 10,11, Shawn T. Sullivan1 & Sarah B. Kingan 2✉

Haplotype-resolved genome assemblies are important for understanding how combinations of variants impact phenotypes. To date, these assemblies have been best created with complex protocols, such as cultured cells that contain a single-haplotype (haploid) genome, single cells where haplotypes are separated, or co-sequencing of parental genomes in a trio-based approach. These approaches are impractical in most situations. To address this issue, we present FALCON-Phase, a phasing tool that uses ultra-long-range Hi-C chromatin interaction data to extend phase blocks of partially-phased diploid assembles to chromosome or scaffold scale. FALCON-Phase uses the inherent phasing information in Hi-C reads, skipping variant calling, and reduces the computational complexity of phasing. Our method is validated on three benchmark datasets generated as part of the Vertebrate Genomes Project (VGP), including human, cow, and zebra finch, for which high-quality, fully haplotype-resolved assemblies are available using the trio-based approach. FALCON-Phase is accurate without having parental data and performance is better in samples with higher hetero-zygosity. For cow and zebra finch the accuracy is 97% compared to 80–91% for human. FALCON-Phase is applicable to any draft assembly that contains long primary contigs and phased associate contigs.

1 Phase Genomics, Seattle, WA, USA. 2 Pacific Biosciences, Menlo Park, CA, USA. 3 Genome Informatics Section, Computational and Statistical Genomics Branch, National Human Genome Research Institute, Bethesda, MD, USA. 4 Department of Genome Sciences, University of Washington School of Medicine, Seattle, WA, USA. 5 US Meat Animal Research Center, ARS USDA, Clay Center, NE, USA. 6 Davies Research Centre, School of Animal and Veterinary Sciences, The University of Adelaide, Roseworthy, SA, Australia. 7 Vertebrate Genomes Laboratory, The Rockefeller University, New York, NY, USA. 8 Howard Hughes Medical Institute, University of Washington, Seattle, WA, USA. 9 Dipartimento di Scienze Animali, della Nutrizione e degli Alimenti, Università Cattolica del Sacro Cuore, 29122 Piacenza, Italy. 10 Laboratory of Neurogenetics of Language, The Rockefeller University, New York, NY, USA. 11 Howard Hughes Medical Institute, Chevy Chase, MD, USA. ✉email: zkronenberg@pacificbiosciences.com; skingan@pacificbiosciences.com

High-quality reference genomes are an indispensable resource for basic and applied research in biology, genomics, agriculture, medicine, and many other fields[1–3]. Technological innovations in DNA sequencing, long-range genotyping, and assembly algorithms have led to rapidly declining costs of sequencing and computation for genome assembly projects[4]. A major challenge for de novo assembly of genomes of outbred, non-model, diploid and polyploid organisms is accurate haplotype resolution. Most genome assemblers collapse multiple haplotypes into a single consensus sequence to generate a pseudo-haploid reference. Unfortunately, this process results in mosaic haplotypes with erroneously associated variants not present in either haplotype, with concomitant negative impacts on biological inference[5–7].

Four approaches to haplotype resolution in long-read diploid genome assembly have been described. Trio binning uses short-read sequence data of the parents to identify parent-specific *k*-mers, which are then used to bin long-read sequence data of the offspring into maternal and paternal bins[8–10]. These parent-specific read bins can be separately assembled into two haploid genomes, as with TrioCanu[9] or binned within the assembly graph, as with hifiasm[10]. Trio binning provides accurate phased assemblies but requires that samples of the parents are available, which is often not possible. A second approach phases reads by mapping to an existing reference genome to infer haplotypes, followed by long-read partitioning and assembly[11–14]. Read-based phasing methods require that a reference assembly is available and depends on single-nucleotide variant (SNV) calling, which has associated errors. A third approach is to use Strand-seq[15] to sequence DNA template strands only, but not the nascent strands that have been selectively labeled and targeted for removal. The advantage of this method is that structural contiguity of individual homologs is maintained, but it requires living cells and at least one cell division with BrdU labeling, and thus is not easily scalable for many species or individuals of a species. The fourth approach is to separate haplotypes during the genome assembly process as implemented by FALCON-Unzip for long reads[16], DipAsm for Hi-C and long reads[17], and Supernova for short reads[18]. The length of the phase blocks produced by these methods are, however, limited by sequence read length and depth of coverage in the diploid genome.

To address these issues, we developed FALCON-Phase, an assembly processing pipeline that uses the natural intra-chromosomal interactions identified by Hi-C to phase paternal and maternal contigs and their associated haplotigs from a long-read assembly of a diploid organism. A haplotig is an assembled sequence from a single haplotype and there are typically several haplotigs interspersed along their primary contig (Fig. 1). A fundamental limitation of partially phased long-read assemblies is that the phase between neighboring haplotigs is unknown. FALCON-Phase solves this problem in an efficient fashion, not by calling or phasing SNP variants relative to an existing reference genome, but by using the ultra-long-range (>1 Mb) information from the mapping of unique, haplotype-specific, Hi-C read pairs[19–21] and a stochastic algorithm to establish correct linkage between haplotigs along a contigs.

FALCON-Phase uses a partially phased contig assembly and Hi-C data, which can be obtained for many samples, including field-collected organisms for which trio samples may not be available. We apply our method to PacBio long-read de novo genome assemblies of three species with different levels of heterozygosity. Performance of our method is best with high heterozygosity samples: zebra finch (*Taeniopygia guttata*), and an intersubspecies cross of *Bos taurus*[8,9] (a male fetus, but referred to as cow for simplicity), achieving 97% accuracy, whereas the lower-heterozygosity human samples have phasing accuracy of 80–91%. By applying our phasing method to contigs and scaffolds in two separate iterations it is possible to extend haplotype phasing to chromosomes scale.

## Results

**FALCON-Phase: a Hi-C haplotype-phasing tool for long-read assemblies.** FALCON-Phase inputs a partially phased long-read assembly, such as one from FALCON-Unzip, and extends the phasing on the contigs using Hi-C reads from the same sample. The method leverages the higher density of *cis*-interactions for Hi-C read pairs to regroup phase blocks (haplotigs) into haplotypes along a contig[11]. First, the haplotype phase blocks are defined by aligning the alternate haplotigs to their associated primary contigs (Fig. 1b). Breaks (minces) in the contigs are introduced to separate phased from unphased (collapsed haplotype) regions (Fig. 1c). Hi-C read pair mapping density is then used to classify haplotype blocks that are in the same phase (same parental homolog) along each contig (Fig. 1d, e). The assembly sequences are then expanded by integrating the collapsed sequences into both haplotypes to obtain two contig sets that contain either maternal or paternal phase blocks interspersed with the collapsed regions (Fig. 1f). Although FALCON-Phase groups maternal and paternal sequences from the same chromosomes, it is agnostic as to which parent the assembled chromosomes came from. Details of the method, including equations and algorithms, are described in the methods.

**Over 90% of paternal and maternal contigs correctly phased.** We tested FALCON-Phase on three vertebrate species for which we had trio-binned assemblies from the same data: two human samples (HG00733 and mHomSap3), zebra finch, and cow (see "Data availability"). In order to most accurately assess the performance of our method, we removed errors in the starting de novo assembly first by breaking chimeric contigs containing sequences from different chromosomes for all samples using visualization of Hi-C read density with Juicebox[22]. Second, for the highest heterozygosity sample, zebra finch, it was also necessary to run purge haplotigs[23] to remove haplotype duplications in the primary contig set. After this assembly curation, the primary contig assemblies ranged from ~1 to 3 Gb in size, matching the expected haploid genome size, with contig N50 values from ~3 to 30 Mb in length and 81–88% of the genomes present in phased haplotigs (Table 1 and Supplementary Table 1). Average alternate haplotig assembly length, which is equivalent to average phase block size, ranged from 188 to 452 kb (Table 1).

In the next stage, Hi-C read pairs were aligned to both the collapsed regions and phase blocks using the software BWA-MEM[24]. By requiring both Hi-C read pairs to have a map quality greater than 10, we obtained a haplotype-specific set of Hi-C reads. We found that depending on sample heterozygosity level (Table 1), between ~11 and 44% of the Hi-C read pairs contained haplotype-specific variants (Supplementary Table 2). A matrix was then generated from the counts of retained Hi-C read pairs mapping between phase blocks, and the phasing algorithm was then applied. We assessed phasing performance of our method by counting parental *k*-mers identified in Illumina sequence data from the parents and used a stringent measure that penalized every *k*-mer that was contained within an erroneous phase switch. We ran FALCON-Phase on 64 CPUs, with 488GB RAM, and a 600GB magnetic disk. For the mHomSap3 dataset the total wall time was 46 h and the total CPU time was 579 h. The majority of time was spent mapping the HiC data (549 CPU hours) and running the phasing algorithm (25 CPU hours).

Before applying FALCON-Phase, ~61–75% of the primary contig *k*-mers and ~95–98% of the haplotig *k*-mers were

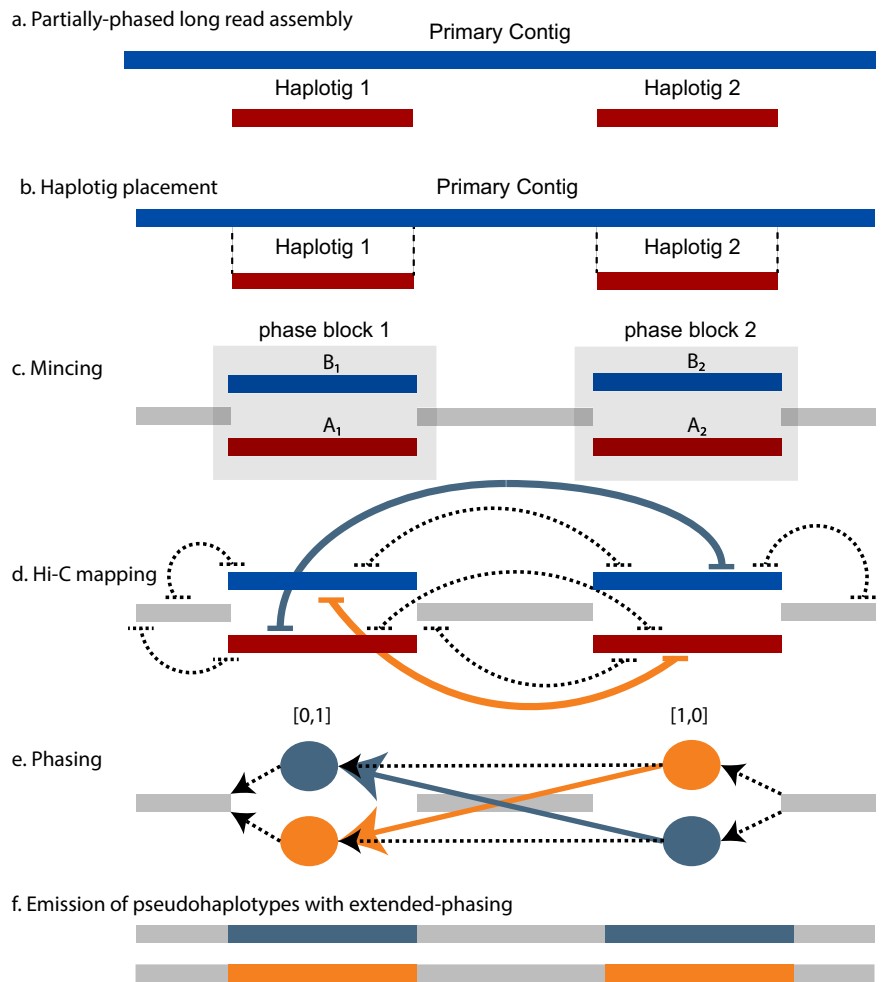

**Fig. 1 Overview of FALCON-Phase method. a** Partially phased long-read assembly consists of primary contigs (blue) and shorter alternate haplotigs (red). The region where a haplotig overlaps a primary contig is a phase block and is referred to as being unzipped because two haplotypes are resolved. Regions of the primary contig without associated haplotigs are referred to as collapsed because the haplotypes have low or no heterozygosity. **b** A haplotig placement file specifies primary contig coordinates where the haplotigs align. **c** This placement file is used to mince the primary contigs at the haplotig alignment start and end coordinates. Mincing defines the phase blocks (A–B haplotype pairs, blue and red) and collapsed haplotypes (gray). **d** Hi-C read pairs are mapped to the minced contigs and alignments are filtered to retain haplotype-specific mapping. **e** Phase blocks are assigned to state 0 or 1 using the phasing algorithm. **f** The output of FALCON-Phase is two full-length pseudo-haplotypes for phase 0 and 1. These sequences are of similar length to the original primary contig and the unzipped haplotypes are in phase with each other.

**Table 1 Input statistics for the genomes used for FALCON-Phase.**

| Sample heterozygosity | | | | |
|---|---|---|---|---|
| **Sample** | **Zebra finch** | **Cow** | **HG00733** | **mHomSap3** |
| Heterozygosity (%) | 1.57–1.72 | 0.65–0.93 | 0.17–0.21 | 0.25–0.26 |
| **Contig and Hi-C summary statistics** | | | | |
| Primary assembly length (Gb) | 1.05 | 2.71 | 2.89 | 2.88 |
| Primary Contig N50 (Mb) | 3.48 | 31.4 | 26.3 | 22.4 |
| Mean Phase Block Length (kb) | 188 | 452 | 312 | 351 |
| Proportion of genome unzipped (%) | 87.6 | 87.7 | 84.0 | 81.1 |
| Average number of Hi-C links between phase blocks on the same primary contig (pre/post) filtering | 92.5/31.5 | 20.39/4.79 | 44.79/2.42 | 16.28/10.10 |
| **Scaffold summary statistics** | | | | |
| Number of scaffolds | 30 | 31 | 23 | 28 |
| Total scaffold length (Gb) | 1.06 | 2.64 | 2.86 | 2.87 |
| Number of gaps | 740 | 962 | 523 | 850 |
| Number of contigs scaffolded | 797 | 1040 | 514 | 862 |
| Number of unscaffolded contigs | 160 | 650 | 351 | 207 |

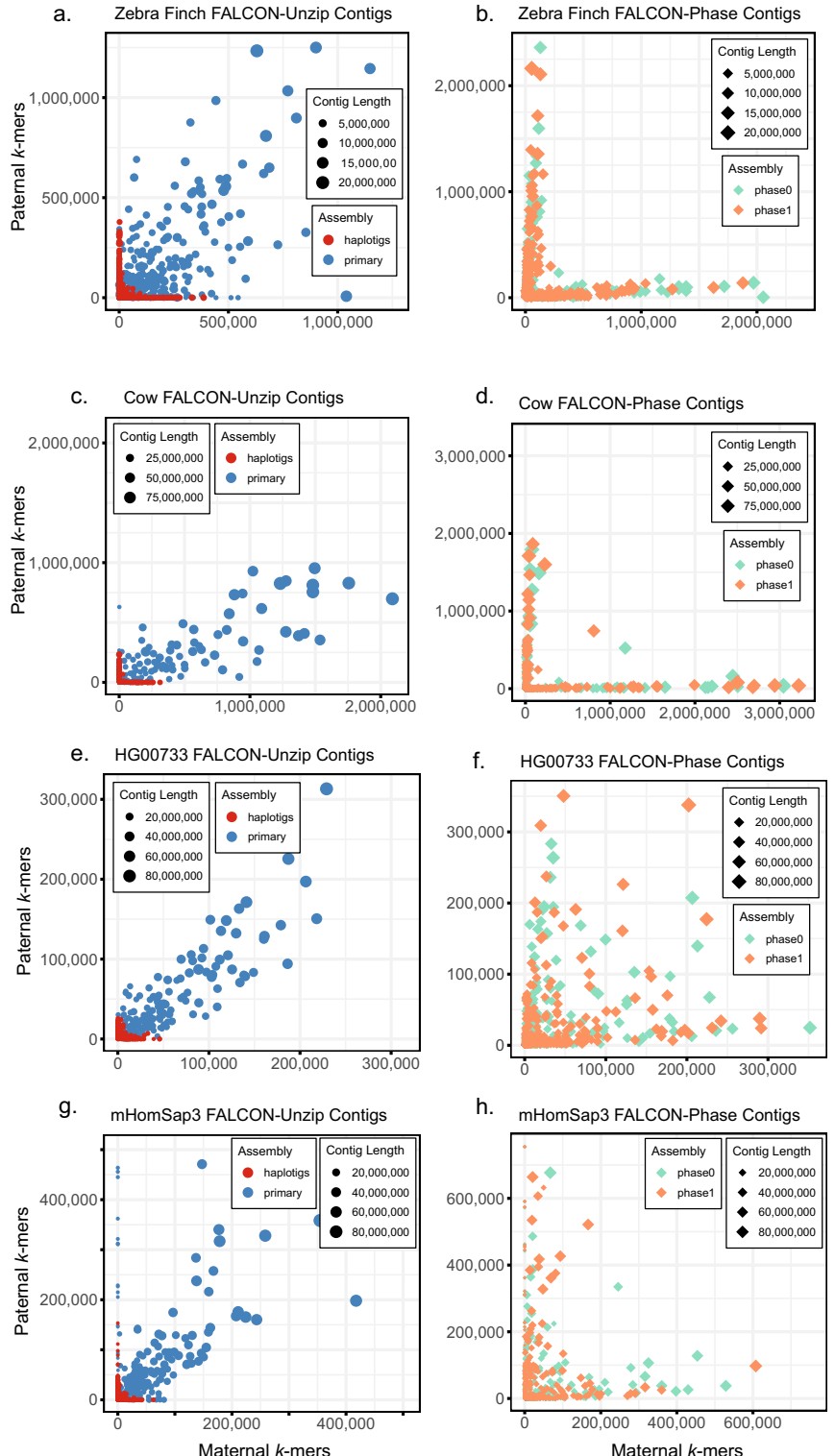

**Fig. 2 Phasing accuracy of contigs before (left) and after applying FALCON-Phase (right) to the contigs.** Parent-specific *k*-mer count from mother is on the *x*-axis and father on the *y*-axis. Contig size is indicated by size of the data point and well-phased contigs lie along the axes. Unphased primary contigs (blue) are large but contain a mixture of *k*-mer markers from mother and father. Haplotigs are mostly phased but shorter in length. After phasing by FALCON-Phase, phase 0 and phase 1 contigs are of similar length to the FALCON-Unzip primary contigs and have less mixing of parental markers within contigs. **a** Zebra finch contigs before phasing; **b** zebra finch contigs after phasing; **c** cow contigs before phasing; **d** cow contigs after phasing; **e** HG00733 contigs before phasing; **f** HG00733 contigs after phasing; **g** mHomSap4 contigs before phasing; **h** mHomSap3 contigs after phasing.

**Table 2 FALCON-Phase performance.**

| Contig phasing accuracy | | | | |
| --- | --- | --- | --- | --- |
| Sample | Zebra finch | Cow | HG00733 | mHomSap3 |
| FALCON-Unzip Primary Contig Accuracy (%) | 70.8 | 71.0 | 61.0 | 75.5 |
| FALCON-Unzip Haplotig Accuracy (%) | 94.9 | 98.7 | 96.2 | 98.3 |
| FALCON-Phase Contig Accuracy (%) | 91.2 | 96.0 | 80.3 | 91.2 |
| Trio-binned Canu Contig Accuracy (%) | 99.4 | 99.4 | 99.5 | 99.6 |
| **Scaffold phasing accuracy** | | | | |
| Unphased Scaffold Accuracy (%) | 64.1 | 77.8 | 62.9 | 75.7 |
| FALCON-Phase Scaffold Accuracy (%) | 88.4 | 92.4 | 73.9 | 84.9 |

accurately phased into their paternal or maternal haplotypes (Fig. 2a, c, e, g and Table 2; see also ref. [25]). After applying FALCON-Phase, the accuracy of the phasing of the new contigs was 91–96% for cow, zebra finch, and mHomSap3 (Fig. 2b, d, f, h and Table 2). The accuracy for the HG00733 human was lower at 80.3%, likely due to poor quality Hi-C data (see below for more detail). In comparison, trio-binned Canu assemblies have >99% parental phasing accuracy for these genomes. We also evaluated the phase accuracy of a supernova assembly of the HG00733 sample and determined it to be 74% for parental haplotypes (Supplementary Fig. 1). We also applied FALCON-Phase to a PacBio HiFi assembly of HG002 and saw similar performance to the other humans (Supplementary Table 3).

The FALCON-Unzip assemblies of the two human samples had similar contiguity (primary contig N50 = 22.4 for mHomSap3 and 26.3 Mb for HG00733), mean phase block length (0.351 Mb for mHomSap3 and 0.312 Mb for HG00733), and percent of the genome unzipped (81% for mHomSap3 and 84% for HG00733; Table 1), although the heterozygosity for mHomSap3 is slightly higher than for HG00733 (0.26% versus 0.21%). Interestingly, both the absolute number and percentage of long-range Hi-C contacts for mHomSap3 are much higher than that of HG00733: 12M versus 4.5M Hi-C read pairs have mapping distance greater than 100 kb (6.6% versus 3.5% of filtered reads have >100 kb mapping distance, Supplementary Table 3 and Supplementary Fig. 2). A possible explanation for the poorer Hi-C data of HG00733 is that it was collected from a frozen cell line whereas the mHomSap3 Hi-C data were collected from fresh blood.

**Over 85% of paternal and maternal scaffolds correctly phased.** One set of the resulting contigs from FALCON-Phase (phase 0) was scaffolded into chromosome-scale sequences using Proximo Hi-C (Phase Genomics, Table 1 and Fig. 3). A second round of phasing was performed on the scaffolds using FALCON-Phase and performance was evaluated using parental $k$-mer counts in the unphased versus phased scaffolds (Table 2). We compare the phasing accuracy of the scaffolds before running FALCON-Phase as a baseline to assess performance for the second round of phasing. In the non-human samples, the unphased scaffolds had between ~62% (zebra finch) and ~78% (cow) phasing accuracy (Table 2); after the second round of FALCON-Phase, accuracy increased to ~88% and ~92%, respectively (Table 2). For the human samples, unphased scaffolds had ~63% (HG00733) and ~78% (mHomSap3) phasing accuracy. Phasing performance in mHomSap3 was good (85% accuracy), compared to HG00733 (74%), which had similarly bad performance for contig phasing due to the poor quality of the Hi-C data (see above). It is important to note that, unlike trio binning, additional information is necessary to compile the maternal or paternal scaffold sets as the phase 0 and phase 1 scaffolds are a mix of maternal and

paternal scaffolds. Also, sex chromosomes and other hemizygous sequence should be treated separately from autosomes.

To independently verify the parental phasing and structural correctness of our human scaffolds, we compared FALCON-Phase HG0733 scaffolds to Strand-seq data from the same individual. Only a small fraction of total length of FALCON-Phase scaffolds genotyped discordantly as homozygous (~0.6%) or heterozygous (~1.6%) (Supplementary Fig. 3). There were 10 putative misassembles at the contig level, which is a commonly observed number for FALCON- or Peregrine-based[26] assemblies when compared to Strand-seq data[27]. The scaffolds had a phasing switch error rate of 0.78 and a hamming distance of 36% (Supplementary Table 4). The hamming distance reported correlates well with the 74% phasing accuracy measured by our $k$-mer counting approach for HG00733. Unfortunately, Strand-seq data were not available for the samples with high-quality Hi-C data so we could not assess them in the same way.

We also explored the performance of our method in the highly heterozygous and repetitive major histocompatibility complex (MHC) region in the mHomSap3 dataset. We identified haplotype phase blocks using Merqury[28] in the chromosome 6 scaffold before and after running FALCON-Phase (Supplementary Fig. 4). Phase blocks were large in the unphased scaffolds: two phase blocks spanned the 4 Mb region around the MHC with a switch between paternal and maternal haplotype near the C4A gene. FALCON-Phase corrected this phase switch, and the final sequence contained only a short segment of paternal haplotype (50 kb) in an otherwise maternal phase block. This phasing error overlaps a putative structural error in our assembly, nested in an array of segmental duplications with greater than 99% sequence identity (Supplementary Fig. 4). Additional orthogonal data are necessary to resolve the discrepancy between our assembly and the hg38 reference.

## Discussion

The ultimate goal of genome assembly is to faithfully represent each chromosome in the organism from telomere-to-telomere. To do so, assembly methods must account for sequence divergence between homologous maternal and paternal chromosomes in order to prevent collapsed haplotypes and false sequence duplications, which may result in incomplete or erroneous representations of the underlying biological sequence[7,9,29]. Long-read genome assemblers like FALCON-Unzip identify heterozygous regions of a genome as bubbles in assembly graphs and unzip those bubbles further by phasing and reassembling reads using single-nucleotide variants (SNVs)[16]. However, long-read assemblers cannot phase entire primary contigs. To address this limitation, we designed FALCON-Phase, which uses Hi-C data to extend the phase blocks to the contig and scaffold scales. Here, we have demonstrated that FALCON-Phase improves accuracy for heterozygous diploid genome assemblies, without the need for parental, population, or Strand-seq data.

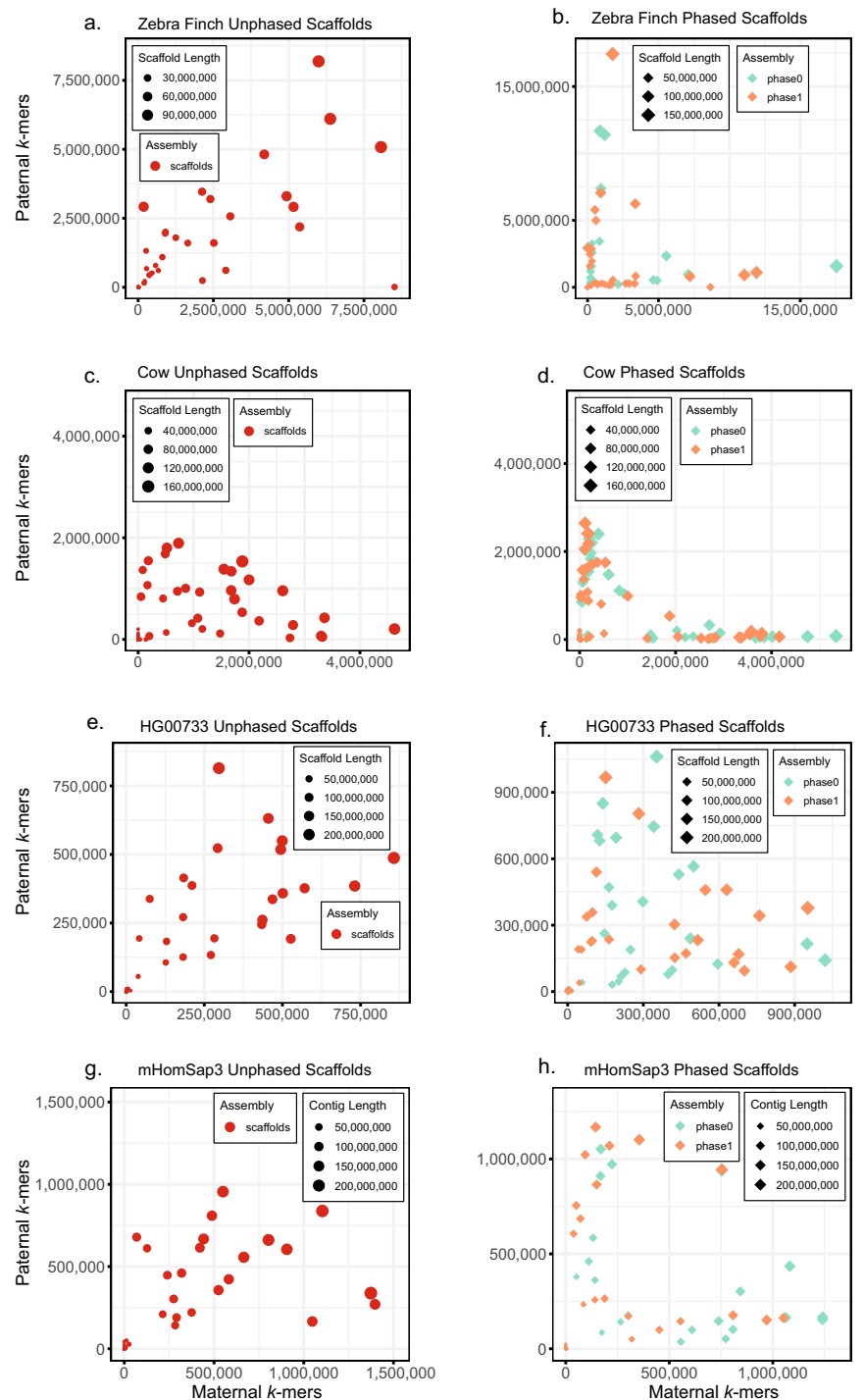

**Fig. 3 Phasing accuracy of scaffolds before (left) and after applying FALCON-Phase (right).** Parent-specific *k*-mers from mother are on the *x*-axis and father on the *y*-axis. Scaffold size is indicated by size of the data point and well-phased contigs lie along the axes. Only the phase 0 contigs from FALCON-Phase were scaffolded. Scaffolds after a second round of phasing by FALCON-Phase show greater separation, indicating each scaffold contains a higher proportion of markers from one parent. **a** Zebra finch scaffolds before phasing; **b** zebra finch scaffolds after phasing; **c** cow scaffolds before phasing; **d** cow scaffolds after phasing; **e** HG00733 scaffolds before phasing; **f** HG00733 scaffolds after phasing; **g** mHomSap4 scaffolds before phasing; **h** mHomSap3 scaffolds after phasing.

FALCON-Phase, in conjunction with long-read assembly, is thus an attractive method for generating high-quality reference genomes of samples for which parents are not available. This approach should be useful for large-scale genome initiatives that source samples of diverse origins, including invertebrate disease vectors, agricultural pests, or threatened or endangered wild-caught individuals. The method utilizes two technologies

common in generating highly contiguous genome assemblies: PacBio long reads and Hi-C. While Hi-C is commonly used for scaffolding[30,31], our study finds that similar high-quality data can also be used for contig or scaffold phasing. The accuracy of phasing increases with Hi-C data quality, specifically the proportion of long-range contacts greater than 100-kb. Coverage requirements of Hi-C for phasing are similar to scaffolding, 100

M reads per Gb of genome size and coverage recommendations for PacBio long reads is at least 60-fold coverage and for PacBio HiFi reads 30-fold coverage. A feature of FALCON-Phase is that it can also be applied to scaffolds in order to link phased scaffold regions. Thus, we suggest the following genome assembly work-flow: (1) partially phased long-read assembly, (2) FALCON-Phase on primary contigs and haplotigs, (3) scaffolding with Hi-C data, and (4) FALCON-Phase on scaffolds.

FALCON-Phase relies on a diploid assembly that is curated as a haploid set of primary contigs plus alternate haplotigs that are each assigned to a primary contig. Generating a high-quality assembly requires the removal of chimeric contigs that join unlinked loci[22,31] in the primary assembly using tools, such as purge haplotigs[32], or purge_dups[33]. Any primary contig is treated as if it were diploid and will be duplicated in the pseudo-haplotype output. Contigs from hemizygous regions of the gen-ome, such as the non-pseudoautosomal regions of sex chromo-somes and mitochondrial sequences (i.e., haploid), cannot have phase-switch errors and should be removed prior to running FALCON-Phase or they will be duplicated as an artifact of the method.

The phasing algorithm at the core of FALCON-Phase could be adapted to use other long-range contact data types and higher ploidies. The input matrix is simply a count of contacts between all pairs of sequences in an assembly. Instead of Hi-C data, BAC-end sequences, read clouds/linked-reads, or optical maps could be transformed into the required input for FALCON-Phase. Hi-C was chosen over the other technologies because it provides ultra-range contact information (>1 Mb), which enables chromosome-scale phase blocks to be created. Similarly, the input sequences could consist of phase blocks generated through resequencing and variant calling, or pseudo-haplotypes generated from assemblies of PacBio HiFi reads or Oxford Nanopore reads (see Supple-mentary Table 3 where we apply the method to a PacBio HiFi assembly of HG002). The simple approach of skirting variant calling reduces the number of steps and overall runtime of phasing pseudo-diploid assemblies. There are additional finishing steps before the assembly is ready for genome annotation, e.g., gap filling with a tool such as PB Jelly[34]. For these reasons, we believe FALCON-Phase will be an important algorithmic con-tribution to the goal of diploid, high-quality genome assemblies.

## Methods

**FALCON-Phase method**. FALCON-Phase has three stages: (1) processing partially phased contigs and Hi-C data; (2) application of the phasing algorithm; and (3) emission of phased pseudo-haplotypes (Fig. 1). We implemented FALCON-Phase using the Snakemake language to provide flexibility and pipeline robustness[35]. The pipeline can be run interactively, on a single computer, or submitted to a cluster job scheduler. The code is open source under a Clear BSD plus attribution license and is available through github (https://github.com/phasegenomics/FALCON-Phase).

In stage one, the contigs are processed to identify phase blocks: regions of the genome that have been unzipped into a maternal and paternal pair of haplotypes. For example, FALCON-Unzip generates contiguous primary contigs representing pseudo-haplotypes and shorter phased alternate haplotigs. A haplotig placement file is generated in the pairwise alignment format[36] that specifies the alignment location of each haplotig on the primary contig (Fig. 1). Briefly, haplotigs are aligned, filtered, and processed with three utilities of the mummer v4 package: *nucmer*, *delta-filter*, and *show-coords*[37]. Sub-alignments for each haplotig are chained in one dimension to find the optimal start and end of the placement using the *coords2hp.py* script. Finally, non-unique haplotig mappings and those fully contained by other haplotigs are removed with *filt_hp.py*.

The haplotig placement file is used to generate three minced FASTA files (Fig. 1), A_haplotigs.fasta, B_haplotigs.fasta, and collapsed_haplotypes.fasta. The A haplotigs are the original haplotigs (red in Fig. 1), the B haplotigs are the corresponding homologous region of the primary contigs (the alternate haplotype, blue in Fig. 1c, d), and the collapsed haplotypes are the unphased or collapsed regions of the assembly (gray in Fig. 1). The pairing of the A and B minced haplotigs in the phase blocks and their order along the primary contig is summarized in an index file, ov_index.txt generated by *primary_contig_index.pl*.

The Hi-C reads are mapped to the minced contigs using BWA-MEM, with the Hi-C option (−5) enabled[24]. The mapped reads are streamed to SAMtools,

removing unmapped, secondary, and supplementary alignments (SAMtools -F 2316)[38]. This operation ensures that each mate-pair only contains two alignment records. In the last step of read processing, a map quality score filter of Q10 (for both reads) is applied, removing reads without haplotype-specific sequence. Additionally, we set an edit distance from the reference of less than 5 for both reads. Both more stringent (60) and relaxed (0) map quality filtering resulted in lower phasing accuracy.

The Hi-C mate-pair counts between minced contigs are enumerated into a contact matrix, $M$. Each element, $M_{i,j}$, in the matrix is later normalized by the number of Hi-C restriction enzyme sites, $z$, in both the $i$th and $j$th minced contigs as shown in Eq. (1). The raw count matrix is encoded into a binary matrix format.

$$\hat{M}_{i,j} := \frac{M_{i,j}}{z_i + z_j} \tag{1}$$

We designed an algorithm to extend phasing between haplotig phase blocks based on Hi-C read pair mapping. The algorithm searches for the optimum set of phase block configurations along a primary contig using a stochastic model. The algorithm is given a list, $C$, of tuples for the phase blocks and their sequential ordering along each primary contig. During initialization, each member of the phase block, except the first, is randomly assigned one of the two possible phase configurations for a diploid organism $\in (\{[0, 1], [1, 0]\})$. The phase assignment is stored in array $T$ where 0 corresponds to phase configuration [0, 1]. The first phase block along the primary contig is always assigned to the phase configuration [0, 1] as its orientation is arbitrary. By fixing the first phase block, the search results are comparable across iterations. Phase blocks are only randomly initialized once before the search begins. The algorithm sweeps along the phase blocks of each primary contig, assigning a phase for the blocks, conditioned on the phase assignment of all previous phase blocks and the Hi-C links between them. The *phaseFreq* function (Eq. 2) calculates the frequency of Hi-C links from the current region, $i$, to all past regions, $j$, that have the same phase, i.e., $T_i = T_j = 1 = [1, 0]$.

$$phaseFreq(i, T, \hat{M}, C) = \frac{\sum_{j=0}^{j<i} \gamma(i,j)*\alpha(i,j)}{\sum_{j=0}^{j<i} \beta(i,j)} \tag{2}$$

The *phaseFreq* function takes the index of the current phase block, $i$, the phase assignment of all regions associated with a given primary contig, array $T$, the normalized Hi-C count matrix, $\hat{M}$, and the $C$ array of the phase block tuples. The gamma function (Eq. 3) determines if two phase blocks have the same phase assignment, $T$, and if so returns 1. The alpha function (Eq. 4) gives the normalized *cis* counts of Hi-C links between a pair of phase blocks whereas the beta function (Eq. 5) returns both the *cis* and *trans* counts, which is a normalizing constant.

$$\gamma(i,j) = \begin{cases} 1, & T[i] = T[j] \\ 0, & T[i] \neq T[j] \end{cases} \tag{3}$$

$$\alpha(i,j,\hat{M},C) = \hat{M}[C[i,0],C[j,1]] + \hat{M}[C[i,1],C[j,0]] \tag{4}$$

$$\beta(i,j,\hat{M},C) = \hat{M}[C[i,0],C[j,0]] + \hat{M}[C[i,1],C[j,1]] + \hat{M}[C[i,0],C[j,1]] + \hat{M}[C[i,1],C[j,0]] \tag{5}$$

The process of phase assignment across a primary contig is iterated for a burn-in period followed by a scoring period (see Algorithm 1). The only difference between the two stages is that the scoring stage enumerates the number of iterations that each member of the phase block spends in phase 1 [1, 0]. We found by ignoring several million iterative sweeps over a primary contig, the algorithm tends to be in a more favorable search space. The final phase assignment is the configuration in which each member of a phase block spent the most iterations. In practice, 50–100 M iterations with 10 M burn-in period generated consistent results. The limiting computational resource is memory as ($\hat{M}$) is not sparse.

## Algorithm 1.

Phasing procedure

**Data**: *normalized HiC count matrix ($\hat{M}$), contig overlap index array (C), number of permutations (n) and burn in (b)*
**Result**: *(R) array, the phase of the A–B haplotig pairs is ε {0,1}*
*m ← length of C−1*
*R ≡ result array of length of C*
*T ≡ temporary phase array of length of C*
*P ≡ state count array (T[i] = 1) of length C*
**if** *length of C ==1* **then**
  **return** *R[0] ← random (ε {0.1})*
**end**
**for** *j ← 0 **to** m* **do**
  *R[j] ← T[j] ← random (ε{0.1})*
  *P[j] ← 0;*
**end**
**for** *i ← 0 **to** n* **do**
  **for** *j ← 0 **to** m* **do**
    *T[j] ← 1;*
    **if** *phaseFreq (j, T, $\hat{M}$, C) < runif (☐)* **then**

```
        T[j] ← 0;
    end
    if i > band T[j] = 1 then
        P[j] ← P[j] + 1
    end
  end
end
for j ← 0 to m do
    R[j] ← 1;
    if P[j]/(n−b) < 0.5 then
        R[j] ← 0;
    end
end
Return R
```

Once the phase assignments of haplotype pairs in the phase blocks are determined, the minced fasta sequences are joined into two full-length pseudo-haplotypes (phase 0 and phase 1) per primary contig (Fig. 1). The order of minced sequences (phase blocks plus collapsed regions) is determined by the haplotig placement file and the phase assignment is determined by the phasing algorithm. An alternate output similar to the FALCON-Unzip format of primary contigs and haplotigs is also available as a user-specified option. Users can specify pseudo-haplotype or unzip output formats, the former having the same collapsed sequence in both pseudo-haplotypes, the latter matching the FALCON-Unzip assembly output format (primary contigs plus haplotigs).

We scaffolded the contigs from FALCON-Phase for the non-human datasets using default Proximo[30,39] settings (Phase Genomics, WA). Briefly, reads were aligned to phase 0 pseudo-haplotypes using BWA-MEM[40] (v. 0.7.15-r1144-dirty) with the -5SP and -t 8 options. SAMBLASTER[41] (commit 37142b37e4f0026e1b83ca3f1545d1807ef77617) was used to flag PCR duplicates, which were later excluded from analysis. Alignments were then filtered with SAMtools (v1.5, with htslib 1.5) using the -F 2304 filtering flag to remove non-primary and supplementary alignments, as well as read pairs in which one or more mates were unmapped. The Phase Genomics Proximo Hi-C genome scaffolding platform (commit 145c01be162be85c060c567d576bb4786496c032) was used to create chromosome-scale scaffolds from the draft assembly as previously described[39]. As in the LACHESIS method[30], this process computes a contact frequency matrix from the aligned Hi-C read pairs, normalized by the number of restriction sites on each contig, and constructs scaffolds in such a way as to optimize expected contact frequency and other statistical patterns in Hi-C data. Juicebox v1.8.8 was used to correct scaffolding errors[22,42]. After scaffolding, we applied the phasing algorithm a second time, using as input the pairing of the phase 0 and phase 1 pseudo-haplotypes and their order along the chromosomes as determined by scaffolding.

We evaluated FALCON-Phase on three vertebrate species with different levels of heterozygosity: The VGP zebra finch female trio (*T. guttata*, high); the male bovine trio (*B. taurus taurus × B. taurus indicus* moderate); Puerto Rican human female trio, (HG00733, low); the VGP admixed human male trio (mHomSap3, low). For each genome, we had high-coverage PacBio CLR data for de novo genome assembly, Hi-C data for phasing and scaffolding, paired-end Illumina data from the parents, and trio-binned Canu assemblies (see "Data availability").

Heterozygosity was estimated two ways. First, from *k-mers* (k-length sequence) in Illumina whole-genome sequencing reads (see "Data availability"). Fastq files were converted to fasta files, then the canonical *k-mers* were collected using meryl in canu 1.7 (ref. [9]) to include all the high frequency *k-mers* using the following code.

```
meryl -B -C -s $name.fa -m $k_size -o $name.$k
meryl -Dh -s $name.$k > $name.$k.hist
```

Given the *k-mer* histogram, Genomescope[43] was used to estimate the level of heterozygosity. $k = 21$ was used for HG00733 and cow, and $k = 31$ was used for the zebra finch and mHomSap3. A higher *k-mer* size was used for zebra finch for more accurate estimates of heterozygosity due to its higher level of polymorphism. This *k-mer* size was also used for other samples in the VGP, from which this sample was selected. Second, with *mummer* v 3.2.3 (ref. [44]), trio-binned parental Canu assemblies were aligned with *nucmer* (nucmer –l 100 -c 500 -maxmatch mom.fasta dad.fasta) and heterozygosity was computed as $1 − \text{average identity from 1 to 1}$ alignments output by *dnadiff* using default parameters.

As a precursor to FALCON-Phase, we performed de novo genome assembly with FALCON-Unzip[16] using pb-assembly from pbbioconda (v 0.0.6 for mHomSap3, v 0.0.2 for zebra finch and cow) and a binary build from13 August 2018, for HG00733.

*Zebra finch parameters*: (length_cutoff = 13,653; length_cutoff_pr = 5000; pa_daligner_option = -e0.76 -l2,000 -k18 -h70 -w8 -s100; ovlp_daligner_option = -k24 -h1024 -e.95 -l1800 -s100; pa_HPCdaligner_option = -v -B128 -M24; ovlp_HPCdaligner_option = -v -B128 -M24; pa_HPCTANmask_option = -k18 -h480 -w8 -e.8 -s100; pa_HPCREPmask_option = -k18 -h480 -w8 -e.8 -s100; pa_DBsplit_option = -x500 -s400; ovlp_DBsplit_option = -s400; falcon_sense_option =−output-multi−min-idt 0.70−min-cov 2−max-n-read 400−n-core 24; overlap_filtering_setting =−max-diff 100−max-cov 150−min-cov 2−n-core 24)

*Cow parameters*: (length_cutoff = 14,850; length_cutoff_pr = 12000; pa_daligner_option = -e0.76 -l1200 -k18 -h480 -w8 -s100; ovlp_daligner_option = -k24 -h480 -e.95 -l1800 -s100; pa_HPCdaligner_option = -v -B128 -M24; ovlp_HPCdaligner_option = -v -B128 -M24; pa_HPCTANmask_option = -k18 -h480 -w8 -e.8 -s100; pa_HPCREPmask_option = -k18 -h480 -w8 -e.8 -s100; pa_DBsplit_option = -x500 -s400; ovlp_DBsplit_option = -s400; falcon_sense_option =−output_multi−min_idt 0.70−min_cov 4−max_n_read 200−n_core 24; overlap_filtering_setting = −max_diff 120−max_cov 120−min_cov 4−n_core 24)

*mHomSap3 parameters*: (length_cutoff = 20,375; length_cutoff_pr = 10,000; pa_daligner_option = -k18 -e0.8 -l1000 -h256 -w8 -s100; ovlp_daligner_option = -k24 -e.92 -l1000 -h1024 -s100; pa_HPCdaligner_option = -v -B128 -M24; ovlp_HPCdaligner_option = -v -B128 -M24; pa_HPCTANmask_option = -k18 -h480 -w8 -e.8 -s100; pa_HPCREPmask_option = -k18 -h480 -w8 -e.8 -s100; pa_DBsplit_option = -x500 -s400; ovlp_DBsplit_option = -s400; falcon_sense_option =−output-multi−min-idt 0.70−min-cov 3−max-n-read 100−n-core 4; falcon_sense_skip_contained = False; overlap_filtering_setting = −max-diff 60−max-cov 60−min-cov 2−n-core 12).

*HG00733 parameters*: (length_cutoff = 5000; length_cutoff_pr = 10,000; pa_daligner_option = -k18 -e0.75 -l1200 -h256 -w8 -s100; ovlp_daligner_option = -k24 -e.92 -l1800 -h600 -s100; pa_HPCdaligner_option = -v -B128 -M24; ovlp_HPCdaligner_option = -v -B128 -M24; pa_HPCTANmask_option = -k18 -h480 -w8 -e.8 -s100; pa_HPCREPmask_option = -k18 -h480 -w8 -e.8 -s100; pa_DBsplit_option = -x500 -s400; ovlp_DBsplit_option = -s400; falcon_sense_option =−output-multi−min-idt 0.70−min-cov 4−max-n-read 200−n-core 8; falcon_sense_skip_contained = False; overlap_filtering_setting =−max-diff 60−max-cov 60−min-cov 1−n-core 12).

We identified and corrected chimeric contigs between nonadjacent genomic regions in HG00733, mHomSap, and cow assemblies using Juicebox Assembly Tools[22] and D-GENIES[45]. We interrogated the concordance of the Hi-C data with the PGA scaffolds visually in JBAT. Off-diagonal signals in the heatmap of Hi-C read density are indicative of contig/scaffolding errors. Human and cow contigs and scaffolds with discordant Hi-C signals were aligned, using *minimap2* with the -x asm5 setting, to the human or cow reference genomes. If the contig/scaffold in question mapped chimerically (inter- or intra-chromosomally) to each genome, they were flagged. We manually broke these contigs between phase blocks and reassociated the haplotigs to the two new contigs.

To remove duplicated haplotypes in the primary contigs from the zebra finch FALCON-Unzip assembly, as suggested for highly heterozygous genomes from the VGP[46], we ran purge haplotigs[23] on zebra finch using default settings and coverage estimates from PacBio subreads mapped to the primary contigs[23]. We recategorized 67.1 Mb of primary contigs as haplotigs ($N = 632$) and 25.4 Mb of repetitive sequences ($N = 329$) were discarded.

To evaluate phase assignment, parent-specific *k-mers* were counted in the pseudo-haplotypes before and after contig phasing, before and after scaffold phasing, and in trio-binned Canu assemblies. Parental *k-mers* were identified using Illumina data from the parents[9] using $k = 21$. Parental *k-mers* were counted in the assemblies using the simple-dump utility from Canu v1.7. The proportion of correct parental *k-mers* was used as an overall measure of contig or scaffold phasing and was plotted for each contig or scaffold in Fig. 2.

To evaluate the structural contiguity of FALCON-Phase scaffolds we aligned available Strand-seq data[47] to the HG00733 scaffolds. We used breakpointR[48] in order to detect regions that are consistently genotyped as "HOM" (majority of reads in minus direction) or "HET" (mixture of plus and minus reads) across all Strand-seq libraries. Regions genotyped as HOM suggest a homozygous inversion or misorientation, while regions genotyped as HET points to either a heterozygous inversion, chimerism, or collapsed repetitive region. Phasing accuracy was evaluated using SNVs detected based on alignments of contig stage assemblies to GRCh38 using minimap2 (version 2.17). We evaluate phasing accuracy of our assemblies in comparison to trio-based phasing for HG00733 (ref. [47]). We compare only SNV positions that are shared between phased assemblies and those from trio-based phasing. Then the switch error rate and Hamming distance were calculated as described in Porubsky et al.[49].

**Reporting summary**. Further information on research design is available in the Nature Research Reporting Summary linked to this article.

## Data availability

Zebra finch PacBio long reads, Hi-C data, parental short-read data, triobinned parental Canu assemblies: [https://vgp.github.io/genomeark/Taeniopygia_guttata/]. FALCON-Unzip contigs: [https://www.ncbi.nlm.nih.gov/bioproject/PRJNA604785], [https://www.ncbi.nlm.nih.gov/bioproject/PRJNA604786]. FALCON-Phase contigs: [https://www.ncbi.nlm.nih.gov/bioproject/PRJNA604789], [https://www.ncbi.nlm.nih.gov/bioproject/PRJNA604788]. FALCON-Phase scaffolds: [https://www.ncbi.nlm.nih.gov/bioproject/PRJNA604793], [https://www.ncbi.nlm.nih.gov/bioproject/PRJNA604794].

Cow PacBio long reads, Hi-C data, parental short-read data, triobinned parental canu assemblies: [https://www.ncbi.nlm.nih.gov/bioproject/PRJNA432857]. FALCON-Unzip contigs: [https://www.ncbi.nlm.nih.gov/bioproject/PRJNA604814], [https://www.ncbi.nlm.nih.gov/bioproject/PRJNA604813]. FALCON-Phase contigs: [https://www.ncbi.nlm.

nih.gov/bioproject/PRJNA604823], [https://www.ncbi.nlm.nih.gov/bioproject/PRJNA604824]. FALCON-Phase scaffolds: [https://www.ncbi.nlm.nih.gov/bioproject/PRJNA604826], [https://www.ncbi.nlm.nih.gov/bioproject/PRJNA604827].

HG00733 PacBio long reads: [https://www.ncbi.nlm.nih.gov/sra/SRR7615963]. Hi-C data: [https://www.ncbi.nlm.nih.gov/sra/ERR1225141], [https://www.ncbi.nlm.nih.gov/sra/ERR1225146]. Parental short-read data: [https://www.ncbi.nlm.nih.gov/bioproject/PRJNA42573]. Triobinned parental canu assemblies: [https://obj.umiacs.umd.edu/marbl_publications/triobinning/h_sapiens_HG00733_dad.fasta], [https://obj.umiacs.umd.edu/marbl_publications/triobinning/h_sapiens_HG00733_mom.fasta]. FALCON-Unzip contigs: [https://www.ncbi.nlm.nih.gov/bioproject/PRJNA604844], [https://www.ncbi.nlm.nih.gov/bioproject/PRJNA604845], [https://www.ncbi.nlm.nih.gov/bioproject/PRJNA604846]. FALCON-Phase contigs: [https://www.ncbi.nlm.nih.gov/bioproject/PRJNA604845], [https://www.ncbi.nlm.nih.gov/bioproject/PRJNA604846]. FALCON-Phase scaffolds: [https://www.ncbi.nlm.nih.gov/assembly/GCA_003634875.1]

mHomSap3 PacBio long reads, Hi-C data, parental short-read data: [https://vgp.github.io/genomeark/Homo_sapiens/]. Triobinned parental canu assemblies: [https://genomeark.s3.amazonaws.com/species/Homo_sapiens/mHomSap3/assembly_nhgri_trio_1.6/intermediates/mHomSap3_mat_t1.fasta.gz], [https://genomeark.s3.amazonaws.com/species/Homo_sapiens/mHomSap3/assembly_nhgri_trio_1.6/intermediates/mHomSap3_pat_t1.fasta.gz]. FALCON-Unzip contigs: [https://www.ncbi.nlm.nih.gov/bioproject/PRJNA604831], [https://www.ncbi.nlm.nih.gov/bioproject/PRJNA604832]. FALCON-Phase contigs: [https://www.ncbi.nlm.nih.gov/bioproject/PRJNA604836], [https://www.ncbi.nlm.nih.gov/bioproject/PRJNA604835]. FALCON-Phase scaffolds: [https://www.ncbi.nlm.nih.gov/bioproject/PRJNA604839], [https://www.ncbi.nlm.nih.gov/bioproject/PRJNA604838].

HG002 PacBio HiFi Reads: [https://www.ncbi.nlm.nih.gov/sra/SRR10382244], [https://www.ncbi.nlm.nih.gov/sra/SRR10382245], [https://www.ncbi.nlm.nih.gov/sra/SRR10382248], [https://www.ncbi.nlm.nih.gov/sra/SRR10382249]. Hi-C data: [https://github.com/human-pangenomics/HG002_Data_Freeze_v1.0]. Parental short-read data: [ftp://ftp-trace.ncbi.nlm.nih.gov/giab/ftp/data/AshkenazimTrio/HG004_NA24143_mother/NIST_Illumina_2x250bps/reads/], [ftp://ftp-trace.ncbi.nlm.nih.gov/giab/ftp/data/AshkenazimTrio/HG003_NA24149_father/NIST_Illumina_2x250bps/reads/]. IPA contigs:[https://www.ncbi.nlm.nih.gov/bioproject/PRJNA667512], [https://www.ncbi.nlm.nih.gov/bioproject/PRJNA667511]. FALCON-Phase contigs: [https://www.ncbi.nlm.nih.gov/bioproject/PRJNA667513], [https://www.ncbi.nlm.nih.gov/bioproject/PRJNA667514].

## Code availability
The FALCON-Phase code is open source and available under The Clear BSD + Attribution License: https://github.com/phasegenomics/FALCON-Phase.

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

## Acknowledgements

We wish to thank Tonia Brown whose efforts greatly improved the clarity of the manuscript. E.D.J. contributions were supported by funds from the Howard Hughes Medical Institute and Rockefeller University. We thank Jason Chin and Mark Chaisson for helpful discussion. The mention of trade names or commercial products in this publication is solely for the purpose of providing specific information and does not imply recommendation or endorsement by the U.S. Department of Agriculture. The USDA is an equal opportunity provider and employer.

## Author contributions

Z.N.K., S.B.K., and S.T.S. conceived and designed the algorithm. P.P., R.J.H., K.M.M., K.K., K.A.M., S.H., O.F., T.P.L.S., E.E.E., I.L., and J.L.W. provided samples or collected data. Z.N.K., S.B.K., G.T.C., S.K., A.R, D.P., S.T.S., and W.Y.L. did data analysis and validation. Z.N.K., S.B.K., A.M.P, E.E.E., E.D.J., J.L.W., T.P.L.S., S.K., D.P., and S.T.S, wrote and revised the manuscript.

## Competing interests

E.E.E. is on the scientific advisory board (SAB) of DNAnexus, Inc. [and was an SAB member of Pacific Biosciences, Inc. (2009–2013)]. S.B.K., Z.N.K., P.P., G.T.C., and R.J.H. are employees and share holders of Pacific Biosciences, a company developing single-molecule sequencing technologies. S.T.S. and I.L. are employee and share holders, and Z.N.K. and K.A.M. are shareholder of Phase Genomics, a provider of services and products for Hi-C and other proximity-ligation methods. The remaining authors declare no competing interests.
