## [Peer Review File · Nature Communications]

REVIEWERS' COMMENTS:

Reviewer #1 (Remarks to the Author):

The authors of the manuscript "Extended haplotype-phasing of long-read de novo genome assemblies using Hi-C" did a great job responding to my questions of the previous review that has now been transferred to Nature Communication. As I said before a phased de novo assembly is indeed an important subject and many research efforts are currently dedicated to obtaining one for diploid genomes. Since this is the 2nd review round I only have a few recommendations:

1. I am a bit concerned about the reproducibility and usability since you describe a lot of hands-on work on the result section 2 ("Over 90% of paternal and maternal contigs correctly phased"). It would be good to make clearer what can be achieved based on automatic running the pipeline vs. high expert mode where you tune/correct certain stages. Also is there documentation about what users should look for or tune in this regard?
2. Maybe explain why you are using the unphased scaffolds as a quality metric. This is somehow counter-intuitive, but I assume you want to show that these were hard or should not have been phased? (Line 150+)
3. I frankly do not agree to present the runtime in the methods section. The methods section is there to explain the method and not to present runtime results. I also don't see the point of reporting the wall time only. Readers will have a hard time to estimate if they can run Falcon-Phase on their cluster or not. Thus, I would encourage to report the runtime section in the results (like every other method that is being published).
4. What coverage or other specifications do you recommend for the HiC library? I think that should be outlined somewhere. In addition what coverage Pacbio data do you recommend (I assume HiFi?)

Reviewer #2 (Remarks to the Author):

This is a revised manuscript that develops a computational approach to perform haplotype phasing on contigs or scaffolds, using ultralong-range Hi-C chromatin interaction data.

In the revised manuscript, they have stressed the point that FALCON-Phase is not an assembly tool per se, but more as a tool that take existing assemblies (long primary contigs and associate haplotigs) and generate haplotype-resolved assembly.

For review 2 comment 2, It seems that in the revised manuscript, the workflow is presented as 1. Long read assembly, 2. FALCON-Phase on primary contigs and haplotigs, 3. Scaffold with Hi-C, 4. FALCON-Phase on scaffolds.

For reviewer 2 comment 3, I understand that the method is not considered as an assembler here, but as a haplotyper that uses HiC data to infer haplotypes for contigs or scaffolds. I mentioned PacBio/Nanopore platforms that use assemblers, but certainly 10X (supernova) also use assemblers, and they all generate primary assembly plus alternative haplotypes, which is what this software can handle by using 10X data. I suggested to use HG002 for very practical reasons: many different types of assemblies are already available on HG002 (from pacbio, illumina, 10X Genomics, etc), and they are not haplotyped; yet since 10X data is available, it should be easy to examine how the performance actually improves using the family information for each of the technical platforms. This yield much more information to users regarding the computational methods and its performance under various scenarios. I totally understand that the authors have analyzed HG002 in another manuscript, but presumably that manuscript has a completely different goal than this one, and I do not see it as a conflict by presenting the results that I asked above, especially given that HG002 family is probably the world's most studied genome in terms of sequencing technology.

For reviewer 2 comment 4, the authors claim that this is the first "trio assembly long read method" and therefore it is quite orthogonal to HG002 which is assembled just by using its own data, but

using multiple different short and long-read platforms, and FALCON-phase can potentially help each platform due to its technology-agnostic nature.

For comment 6, if long-read (pacbio/nanopore) assembly are already available, then they should be used and compared to supernova, before being assessed by FALCON-Phase. The authors' argument "We mentioned supernova because it claims to perform nearly complete phasing" is not convincing as we all know that supernova cannot really perform phasing on human genomes (not even remotely, and not even a high quality assembly with large N50). This is partly why it is discontinued as it does not yield sufficiently strong advantage over competing approaches.

Response to reviewers (final review)

Reviewer #1 (Remarks to the Author):

Reviewer 1 comment: The authors of the manuscript “Extended haplotype-phasing of long-read *de novo* genome assemblies using Hi-C” did a great job responding to my questions of the previous review that has now been transferred to Nature Communication. As I said before a phased *de novo* assembly is indeed an important subject and many research efforts are currently dedicated to obtaining one for diploid genomes. Since this is the 2nd review round I only have a few recommendations:

1. I am a bit concerned about the reproducibility and usability since you describe a lot of hands-on work on the result section 2 (“Over 90% of paternal and maternal contigs correctly phased”). It would be good to make clearer what can be achieved based on automatic running the pipeline vs. high expert mode where you tune/correct certain stages. Also is there documentation about what users should look for or tune in this regard?

Response: The hands on manual curation step we ran was part of the normal curation process that occurs in genome projects, where we broke misassembled contigs and removed duplicate haplotypes. Unfortunately, there is no pipeline that automatically assembles a high-quality genome without additional steps to remove false duplicated haplotypes, miss-joins in scaffolds, and other errors. For example, even the vertebrate genome project’s (VGP) pipeline has manual curation steps and corrections.

Nevertheless, no manual intervention was necessary during the FALCON-Phase stage, which is the focus on this paper. After contig assembly, we performed manual scaffold curation as done in the VGP and others, to best assess FALCON-Phase accuracy by removing upstream errors in the contigs.

We added the following sentence to the results to clarify this point (line 112):

“In order to most accurately assess the performance of our method, we removed errors in the starting *de novo* assembly first by breaking chimeric contigs containing sequences from different chromosomes for all samples using visualization of Hi-C read density with Juicebox²². Second, for the highest heterozygosity sample, zebra finch, it was also necessary to run purge haplotigs²³ to remove haplotype duplications in the primary contig set. After this assembly curation, ...”

Our README (<https://github.com/phasegenomics/FALCON-Phase>) provides the details of how to run FALCON-Phase, but not scaffolding because it is not part of the method.

Reviewer 1 comment: 2. Maybe explain why you are using the unphased scaffolds as

a quality metric. This is somehow counter-intuitive, but I assume you want to show that these were hard or should not have been phased? (Line 150+)

Response: Unphased scaffolds are the baseline for contig phasing accuracy. We have added this sentence to the results to clarify (line 166):

“We compare the phasing accuracy of the scaffolds before running FALCON-Phase as a baseline to assess performance for the second round of phasing.”

Reviewer 1 comment: 3. I frankly do not agree to present the runtime in the methods section. The methods section is there to explain the method and not to present runtime results. I also don't see the point of reporting the wall time only. Readers will have a hard time to estimate if they can run Falcon-Phase on their cluster or not. Thus, I would encourage to report the runtime section in the results (like every other method that is being published).

Response: This is a reasonable request. We have moved the estimates of runtime to the results, on line 135. We also added CPU time, besides wall time. The compute requirements of FALCON-Phase are largest for bwa-mem read mapping, and the phasing algorithm.

Reviewer 1 comment: 4. What coverage or other specifications do you recommend for the Hi-C library? I think that should be outlined somewhere. In addition what coverage PacBio data do you recommend (I assume HiFi?)

Response: We have now added suggested coverage specifications, citing lessons learned in companion projects (Rhie et al 2020 biorxiv VGP paper; Koren et al 2020 HiCanu paper). For Hi-C, it is 100M reads per Gb of genome size; for PacBio CLR, it is 60X coverage; for PacBio HiFi, it is 30X coverage.

We added this sentence to the discussion to clarify this (line 222):

“Coverage requirements of Hi-C for phasing are similar to scaffolding, 100M reads per Gb of genome size and coverage recommendations for PacBio long reads is at least 60-fold coverage and for PacBio HiFi reads 30-fold coverage.”

Reviewer #2 (Remarks to the Author):

Reviewer 2 comment: This is a revised manuscript that develops a computational approach to perform haplotype phasing on contigs or scaffolds, using ultralong-range Hi-C chromatin interaction data.

In the revised manuscript, they have stressed the point that FALCON-Phase is not an assembly tool per se, but more as a tool that take existing assemblies (long primary contigs and associate haplotigs) and generate haplotype-resolved assembly.

For reviewer 2 comment 2, It seems that in the revised manuscript, the workflow is presented as 1. Long read assembly, 2. FALCON-Phase on primary contigs and haplotigs, 3. Scaffold with Hi-C, 4. FALCON-Phase on scaffolds.

Response: We are glad to see that our revisions have made the key points of the paper more transparent. The summary above does not miss a single point.

For reviewer 2 comment 3, I understand that the method is not considered as an assembler here, but as a haplotyper that uses Hi-C data to infer haplotypes for contigs or scaffolds. I mentioned PacBio/Nanopore platforms that use assemblers, but certainly 10X (supernova) also use assemblers, and they all generate primary assembly plus alternative haplotypes, which is what this software can handle by using 10X data. I suggested to use HG002 for very practical reasons: many different types of assemblies are already available on HG002 (from pacbio, illumina, 10X Genomics, etc), and they are not haplotyped; yet since 10X data is available, it should be easy to examine how the performance actually improves using the family information for each of the technical platforms. This yield much more information to users regarding the computational methods and its performance under various scenarios. I totally understand that the authors have analyzed HG002 in another manuscript, but presumably that manuscript has a completely different goal than this one, and I do not see it as a conflict by presenting the results that I asked above, especially given that HG002 family is probably the world's most studied genome in terms of sequencing technology.

Response: The reviewer rightly points out that HG002 is a good sample, as it the best characterized human genome in terms of number of available genomic datasets. We have run FALCON-Phase (at the contig level) on HG002. Unlike the other genomes, we used PacBio HiFi data for the starting contig assembly. The results were consistent with the other human genomes – the method improved the phasing. We verified phasing accuracy using the *k*-mer analysis with Illumina data from HG003 and HG004 (parents). As these results echo the other human genomes we have put these results in the supplementary material (see Supplementary Table 3)

For reviewer 2 comment 4, the authors claim that this is the first "trio assembly long read method" and therefore it is quite orthogonal to HG002 which is assembled just by using its own data, but using multiple different short and long-read platforms, and FALCON-phase can potentially help each platform due to its technology-agnostic nature.

Response: Our method is among the first non-trio phasing methods for long reads that gives a reasonably complete assembly of each haplotype. The published assemblies of HG002 have not yet performed a complete phased assembly without parental data. The human pangenomes project is working on such an assembly, but this is still in development. Yes, FALCON-Phase can be used on multiple platforms. The intent with this study though was not to perform a thorough comparison of different technologies, but rather to validate a new phasing method that can be broadly applied to samples

without family trios. To that end, we have tried to be focused in this study, limiting our primary analysis to two commonly used data types, PacBio and Hi-C data. FALCON-Phase can be extended to be agnostic to the technology used to generate the contigs, and if a user understands the input styles, they are able to supply any starting genome assembly. In the revised paper, we mentioned that to accommodate other data types, such as pair-end linked reads, read mapping and processing could be modified to substitute them for the Hi-C input in FALCON-Phase (see paragraph starting on line 237).

For comment 6, if long-read (pacbio/nanopore) assembly are already available, then they should be used and compared to supernova, before being assessed by FALCON-Phase. The authors' argument "We mentioned supernova because it claims to perform nearly complete phasing" is not convincing as we all know that supernova cannot really perform phasing on human genomes (not even remotely, and not even a high quality assembly with large N50). This is partly why it is discontinued as it does not yield sufficiently strong advantage over competing approaches.

Response: We note that we do compare the HG00733 supernova assembly to the PacBio long read assembly (see Supplementary Figure 2 and main text line 146). However, we think it is beyond the scope of this work to compare the phasing accuracy of additional long read assemblies, as our main goal is to demonstrate a new phasing method that is applied *after* contig-assembly. As the reviewer notes and others have shown (Rhie et al. 2020, biorxiv), 10X linked read technologies produce lower quality assemblies compared to those using long read technologies and the technology has been discontinued. We have instead opted to use Strand-seq data as an orthogonal datatype which has been shown to provide chromosome-scale phasing information (Porubsky et al. 2017).